# The Surprising Effectiveness of Negative Reinforcement in LLM Reasoning

**Xinyu Zhu**[1]  **Mengzhou Xia**[2]  **Zhepei Wei**[1]  **Wei-Lin Chen**[1]
**Danqi Chen**[2]  **Yu Meng**[1]
[1]Computer Science Department, University of Virginia
[2]Princeton Language and Intelligence (PLI), Princeton University
{xinyuzhu,zhepei.wei,wlchen,yumeng5}@virginia.edu
{mengzhou,danqic}@cs.princeton.edu

## Abstract

Reinforcement learning with verifiable rewards (RLVR) is a promising approach for training language models (LMs) on reasoning tasks that elicit emergent long chains of thought (CoTs). Unlike supervised learning, it updates the model using both correct and incorrect samples via policy gradients. To better understand its mechanism, we decompose the learning signal into reinforcing correct responses and penalizing incorrect ones, referred to as **P**ositive and **N**egative **S**ample **R**einforcement (**PSR** and **NSR**), respectively. We train `Qwen2.5-Math-7B`, `Qwen3-4B` and `Llama-3.1-8B-Instruct` on a mathematical reasoning dataset and uncover a surprising result: training with only negative samples—without reinforcing correct responses—can be highly effective: it consistently improves performance over the base model across the entire Pass@$k$ spectrum ($k$ up to 256), often matching or surpassing PPO and GRPO. In contrast, reinforcing only correct responses improves Pass@1 but degrades performance at higher $k$, due to reduced diversity. These inference-scaling trends highlight that solely penalizing incorrect responses may contribute more to performance than previously recognized. Through gradient analysis, we show that NSR works by suppressing incorrect generations and redistributing probability mass toward other plausible candidates, guided by the model's prior beliefs. It refines the model's existing knowledge rather than introducing entirely new behaviors. Building on this insight, we propose a simple variant of the RL objective that upweights NSR, and show that it consistently improves overall Pass@$k$ performance on MATH, AIME 2025, and AMC23. Our code is available at https://github.com/TianHongZXY/RLVR-Decomposed.

## 1 Introduction

Language models (LMs) have recently demonstrated remarkable capabilities in various complex reasoning tasks, including mathematics [7, 17], coding [21, 66], and scientific reasoning [37, 40]. A key technique in achieving such success is reinforcement learning with verifiable rewards (RLVR) [15, 20, 23, 47], which is particularly effective in domains where the correctness of an outcome can be automatically verified via tools or functions. RLVR typically employs a binary reward ($+1$ or $-1$) based on the objective correctness of model responses. This simple yet effective mechanism not only mitigates reward hacking [33, 45] but also eliminates the need for extensive human annotations and complex reward model training [27, 77].

RLVR's appeal is multifaceted: it offers a conceptually simple formulation [23], exhibits notable sample efficiency [12, 28, 52], and enables inference-time scaling behaviors [13, 35, 60, 68, 74]. However, the precise mechanisms driving its effectiveness remain underexplored, particularly how it utilizes correct and incorrect samples. To address this, we decompose RLVR into two learning paradigms: *Positive Sample Reinforcement (PSR)* and *Negative Sample Reinforcement (NSR)*, as

39th Conference on Neural Information Processing Systems (NeurIPS 2025).

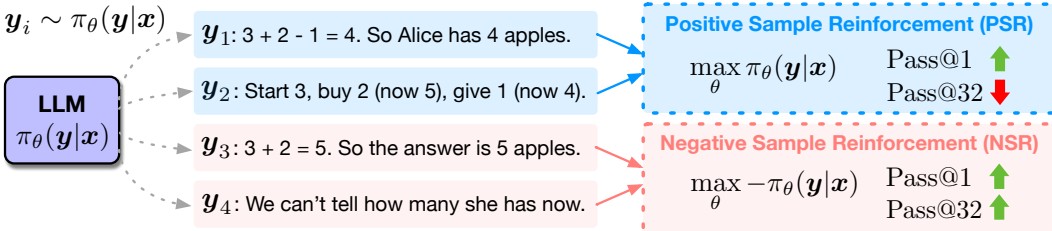

Figure 1: Decomposing learning signals in RLVR into positive and negative reward components. Positive Sample Reinforcement (PSR) increases the likelihood of correct responses and improves Pass@1, but reduces output diversity and hurts Pass@$k$ for large $k$. Negative Sample Reinforcement (NSR) discourages wrong responses and redistributes probability mass according to the model's prior knowledge, improving the full Pass@$k$ spectrum.

illustrated in Figure 1. This decomposition prompts a natural question: What roles do PSR and NSR play in shaping model behavior and generalization?

To empirically isolate the effects of PSR and NSR, we train two LMs, `Qwen2.5-Math-7B` and `Qwen3-4B`, either PSR or NSR exclusively, and evaluate their inference-time performance across a range of Pass@$k$ metrics. We find that PSR-only training improves Pass@1 but hurts Pass@$k$ at larger $k$ values, indicating a loss of output diversity and exploration capacity. More strikingly, NSR-only training consistently improves performance over the base LM across the entire Pass@$k$ spectrum and, in many cases, matches or even surpasses the performance of PPO [41] and GRPO [15, 43].

To understand why NSR alone is so effective, we conduct a token-level gradient analysis and demonstrate that NSR works by suppressing incorrect reasoning steps and redistributing probability mass towards other plausible candidates already favored by the model's prior. This effectively refines its existing knowledge without aggressively teaching new behaviors. PSR, in contrast, sharpens the output distribution around sampled correct paths [1, 16, 61], often at the cost of suppressing alternative valid solutions. This suggests that NSR plays a pivotal role in preserving diversity and promoting generalization, especially when the base model already encodes strong reasoning priors.

Motivated by this insight, we propose Weighted-REINFORCE, a simple yet effective variant of the REINFORCE [58] objective that upweights its NSR contribution. We demonstrate that this adjustment consistently improves the Pass@$k$ performance on MATH, AIME 2025, and AMC23, yielding an overall better result than strong baselines including PPO [41] and GRPO [15, 43].

Our contributions in this work are as follows:

- We decompose RLVR into two components, PSR and NSR, and investigate their distinct impacts on model behavior and generalization measured by a range of Pass@$k$ metrics.

- We empirically demonstrate the surprising effectiveness of NSR-only training and use gradient analysis to show that NSR refines the model's prior by suppressing incorrect reasoning steps and preserving plausible alternatives.

- We propose Weighted-REINFORCE, a simple modification to the RL objective that upweights NSR, yielding consistent gains across complex reasoning benchmarks including MATH, AIME 2025, and AMC23.

## 2 RLVR Objective and Decomposition

### 2.1 Reinforcement Learning with Verifiable Rewards

Reinforcement learning with verifiable rewards (RLVR) has recently emerged as a powerful paradigm for enabling LMs to self-improve on tasks with objectively verifiable outcomes. The reward is given by a deterministic verification function $r$ which assesses whether the model's response $\boldsymbol{y}$ to the prompt $\boldsymbol{x}$ is correct or not. All tokens $\{y_1, y_2, \ldots, y_T\}$ in a response $\boldsymbol{y}$ receive the same reward (*e.g.*, $+1$ for correct responses and $-1$ for incorrect ones).

Formally, given an LM with parameters $\theta$, a set of prompts $\mathcal{D}$ from which $\boldsymbol{x}$ is sampled, and a verifiable reward function $r$, RLVR learns a policy $\pi_\theta$ to minimize the following objective:[1]

$$\mathcal{L}_{\text{RLVR}}(\theta) = -\mathbb{E}_{\boldsymbol{x}\sim\mathcal{D},\boldsymbol{y}\sim\pi_\theta(\cdot|\boldsymbol{x})}[r(\boldsymbol{x},\boldsymbol{y})], \quad r(\boldsymbol{x},\boldsymbol{y}) \in \{-1,+1\}. \tag{1}$$

In common RL algorithms (*e.g.*, PPO [41], GRPO [15, 43]), rewards are further normalized to stabilize training, enforcing a zero mean per batch $\mathcal{B}$ (*i.e.*, $\mathbb{E}_{\boldsymbol{x}\sim\mathcal{B},\boldsymbol{y}\sim\pi_\theta(\cdot|\boldsymbol{x})}[r(\boldsymbol{x},\boldsymbol{y})] = 0$).

## 2.2 Decomposing RLVR into Positive and Negative Sample Reinforcement

While RLVR has demonstrated promising empirical results, its underlying learning dynamics remain underexplored. In particular, it is unclear how the model updates its behavior under this binary outcome reward setting. What the model learns when receiving both positive and negative rewards can be entangled and hard to interpret. To better understand these dynamics, we begin by decomposing the RLVR objective into two distinct learning paradigms: learning from correct responses and learning from incorrect responses. This decomposition allows us to investigate how positive and negative reward signals shape model behavior, which we will show in the next section.

The RLVR objective optimizes the expected reward-weighted likelihood:

$$
\begin{aligned}
\mathcal{L}_{\text{RLVR}}(\theta) &= -\mathbb{E}_{\boldsymbol{x}\sim\mathcal{D}}\left[\sum_{\boldsymbol{y}} r(\boldsymbol{x},\boldsymbol{y})\cdot\pi_\theta(\boldsymbol{y}|\boldsymbol{x})\right], \qquad r(\boldsymbol{x},\boldsymbol{y})\in\{-1,+1\} \\
&= \underbrace{-\mathbb{E}_{\boldsymbol{x}\sim\mathcal{D}}\left[\sum_{\boldsymbol{y}:r(\boldsymbol{x},\boldsymbol{y})=1}\pi_\theta(\boldsymbol{y}|\boldsymbol{x})\right]}_{\mathcal{L}_{\text{PSR}}(\theta)} \underbrace{-\mathbb{E}_{\boldsymbol{x}\sim\mathcal{D}}\left[\sum_{\boldsymbol{y}:r(\boldsymbol{x},\boldsymbol{y})=-1}-\pi_\theta(\boldsymbol{y}|\boldsymbol{x})\right]}_{\mathcal{L}_{\text{NSR}}(\theta)},
\end{aligned}
\tag{2}
$$

where we define two sub-objectives representing each learning paradigm:

$$\mathcal{L}_{\text{PSR}}(\theta) = -\mathbb{E}_{\boldsymbol{x}\sim\mathcal{D}}\left[\sum_{\boldsymbol{y}:r(\boldsymbol{x},\boldsymbol{y})=1}\pi_\theta(\boldsymbol{y}|\boldsymbol{x})\right], \tag{3}$$

$$\mathcal{L}_{\text{NSR}}(\theta) = -\mathbb{E}_{\boldsymbol{x}\sim\mathcal{D}}\left[\sum_{\boldsymbol{y}:r(\boldsymbol{x},\boldsymbol{y})=-1}-\pi_\theta(\boldsymbol{y}|\boldsymbol{x})\right]. \tag{4}$$

We refer to these two learning paradigms as *positive sample reinforcement* (PSR) and *negative sample reinforcement* (NSR). The positive reward case resembles supervised fine-tuning (SFT), where the model is updated to increase the likelihood of correct responses. In contrast, the negative reward case mirrors likelihood minimization that reduces the probability assigned to incorrect responses. Notably, PSR and NSR are on-policy—the responses are sampled from the model itself during training.

Thus, the full RLVR objective decomposes into two sub-objectives $\mathcal{L}_{\text{RLVR}}(\theta) = \mathcal{L}_{\text{PSR}}(\theta) + \mathcal{L}_{\text{NSR}}(\theta)$. This decomposition reveals that RLVR jointly performs PSR on positively rewarded samples and NSR on negatively rewarded ones. To better understand the individual effects of these two learning paradigms on model behavior, we conduct experiments to train LLMs with each sub-objective independently, as well as the full RLVR objective for comparison.

# 3 Positive and Negative Sample Reinforcement for LLM Reasoning

## 3.1 Experimental Setup

**Models.** To understand how different training objectives affect the model's behavior, we train the model with PSR and NSR and evaluate their inference scaling performance on reasoning tasks. Specifically, we use `Qwen2.5-Math-7B` [65], `Qwen3-4B` [64] and `Llama-3.1-8B-Instruct` [11]

---

[1]While regularization mechanisms such as KL penalties and clipping are commonly used in practice to stabilize training, the fundamental learning signal still stems from the verifiable reward. Therefore, to clearly understand how the model learns from success and failure, we base our analysis primarily on the reward learning loss. We discuss in Section 4.3 how our analysis applies to PPO and GRPO.

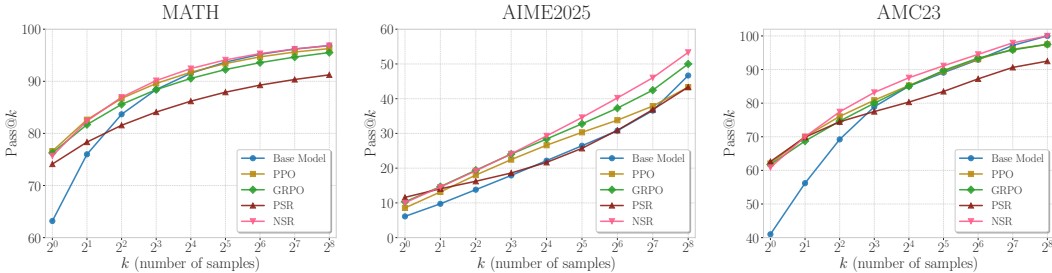

Figure 2: Pass@$k$ curves of `Qwen2.5-Math-7B` trained with PPO, GRPO, PSR, and NSR. NSR is comparable to other methods across different $k$ values and outperforms them at $k = 256$.

as the base models. `Qwen3-4B` has two modes (thinking and non-thinking), and we use the non-thinking mode for training and inference.

**Compared algorithms.** We compare the performance of PSR and NSR with commonly used RL algorithms, including PPO [41] and GRPO [15, 43]. PSR and NSR are implemented by selectively updating the policy model using only correct or incorrect responses, respectively. As a result, PSR and NSR are trained on fewer samples per batch than standard RL algorithms (*e.g.*, PPO and GRPO) that use both correct and incorrect responses. The training objectives of these algorithms can be found in Appendix D.1. We also report the performance of the base models for reference.

**Training setup.** For the training set, we use MATH [17], which contains 7,500 problems. We train the models using the verl framework [44]. The prompt batch size is 1,024, with 8 rollouts generated per prompt. The sampling temperature during training is set to 1.0, and the maximum context length is set to 4,096, 4,096 and 32,768 tokens for `Qwen2.5-Math-7B`, `Llama-3.1-8B-Instruct` and `Qwen3-4B`, respectively. We update the model with a mini-batch size of 256 and a learning rate of 1e-6. More hyperparameter settings can be found in Appendix D.1.

**Evaluation setup.** We evaluate on three widely used math reasoning benchmarks, including the test sets of MATH, AIME 2025 and AMC23. During evaluation, we sample 256 responses per prompt for `Qwen2.5-Math-7B` and `Llama-3.1-8B-Instruct` with a temperature of 0.6 and a top-$p$ of 0.95, and 64 responses for `Qwen3-4B` with a temperature of 0.7, a top-$p$ of 0.8 and a top-$k$ of 20, prompt templates can be found in Appendix D.2. For evaluation metric, recent work [18] highlights that accuracy based on greedy decoding can be unreliable. For this reason, we adopt a full spectrum of Pass@$k$ as our main evaluation metric, using $k \in \{1, 2, 4, 8, 16, 32, 64, 128, 256\}$ for `Qwen2.5-Math-7B` and $k \in \{1, 2, 4, 8, 16, 32, 64\}$ for `Qwen3-4B`. Pass@$k$ is defined as the fraction of problems for which at least one correct response is produced in $k$ independent trials. However, directly computing Pass@$k$ using only $k$ samples per example often suffers from high variance. We follow the unbiased estimator proposed by [5], which generates $n$ samples per problem ($n \geq k$), counts the number of correct responses $c$, and computes an unbiased estimate of Pass@$k$ as:

$$\text{Pass@}k = \mathbb{E}_{\boldsymbol{x} \sim \mathcal{D}} \left[ 1 - \frac{\binom{n-c}{k}}{\binom{n}{k}} \right]. \tag{5}$$

Notably, varying $k$ provides insights into different aspects of model behaviors. Pass@1 approximates greedy decoding accuracy, reflecting how confidently the model can produce a correct response in a single attempt, essentially reflecting *exploitation*. In contrast, Pass@$k$ with large $k$ evaluates the model's ability to generate diverse correct responses across multiple attempts, capturing its *exploration* ability and reasoning boundary.

### 3.2 Inference Scaling Trends Under Different Training Objectives

**NSR alone is surprisingly effective.** As shown in Figures 2 and 3, NSR exhibits unexpectedly strong performance across the full range of $k$ values. Despite being trained solely on negative samples, it consistently improves Pass@$k$ compared to the base model. While PPO, GRPO, and PSR explicitly reinforce correct responses and naturally outperform the base model at Pass@1, it is surprising that NSR achieves a comparable Pass@1 without training on any correct responses. This suggests that

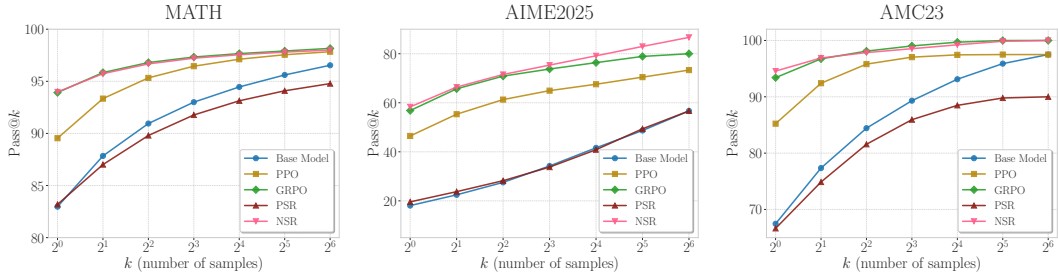

Figure 3: Pass@$k$ curves of `Qwen3-4B` (non-thinking mode) trained with PPO, GRPO, PSR, and NSR. NSR consistently performs competitively across varying $k$ values, while PSR does not improve the base model.

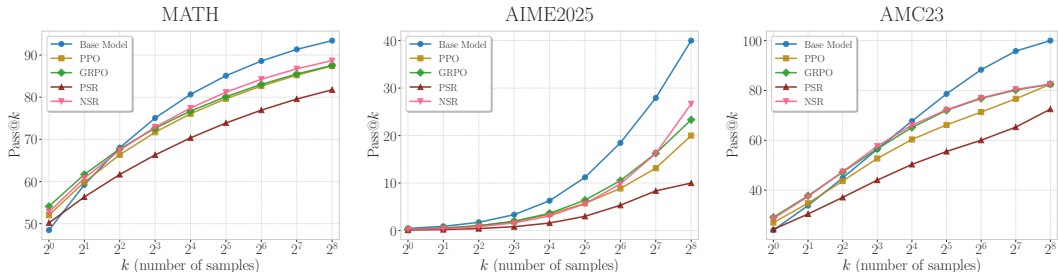

Figure 4: Pass@$k$ curves of `Llama-3.1-8B-Instruct` trained with PPO, GRPO, PSR, and NSR. All the methods underperforms the base model, while NSR retains the most performances.

NSR is able to reinforce correct responses indirectly by suppressing incorrect ones and redistributing probability mass toward plausible alternatives.

**NSR outperforms or stays comparable to the base model at a large $k$ value.**     At larger decoding budgets (*e.g.*, Pass@256), recent work [72] shows that RL-trained models often lose their advantage, and in some cases underperform the base model. This trend is generally observed in our experiments with PPO, GRPO, and PSR, especially in Figure 2. NSR, on the other hand, maintains a comparable or even better Pass@256 performance than the base model, generally outperforming the other algorithms. These results suggest that NSR promotes exploration and preserves the output diversity.

**PSR improves accuracy at the cost of diversity.**     In contrast, PSR, which only reinforces correct samples, displays a more polarized behavior. It improves Pass@1, particularly on AIME 2025 and AMC23 in Figure 2. However, this precision comes at a cost: as $k$ increases, Pass@$k$ improves more slowly than other methods and eventually falls below the base model for $k > 8$. As shown in Figure 5a, PSR improves greedy decoding accuracy rapidly during early training but plateaus quickly and is overtaken by other methods. This behavior indicates that PSR overly concentrates probability mass on early correct responses, leading to overconfidence and a collapsed output distribution and ultimately limiting the model's ability to generate diverse correct responses when allowing for more test-time compute.

**PSR fails to unlock the model's latent reasoning capabilities.**     Figure 3 demonstrates the performance of training `Qwen3-4B` under non-thinking mode to simulate a scenario where the model's underlying knowledge is known to be strong (*i.e.*, its learned thinking mode triggered by the '<think>' tag). While one may intuitively expect all algorithms to easily transfer the model's reasoning ability across different prompt formats (*i.e.*, from thinking to non-thinking mode), our results demonstrate significant performance differences across different algorithms. PSR fails to activate these latent capabilities and does not improve the performance, even degrading Pass@$k$ on MATH and AMC23 significantly. This highlights a fundamental limitation of PSR: it reinforces the currently dominant behavior in the output distribution, suppressing potentially stronger underlying capabilities. In contrast, NSR and GRPO significantly improve the performance of the base model across all Pass@$k$ metrics, achieving thinking mode capabilities. Specifically, `Qwen3-4B` in thinking mode achieves a Pass@1 of 94.5 and a Pass@64 of 97.8 on MATH test set. NSR and GRPO closely match this performance, with NSR reaching 94.0 (Pass@1) and 98.0 (Pass@64), and GRPO achieving 93.9 (Pass@1) and 98.2 (Pass@64).

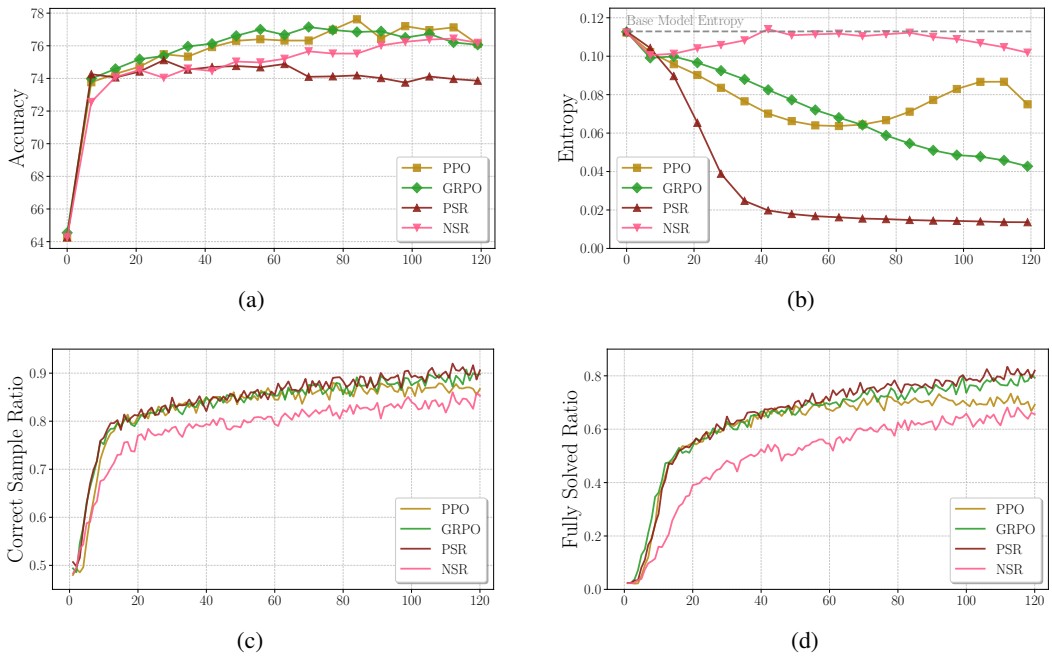

Figure 5: Training dynamics of `Qwen2.5-Math-7B` on MATH under PPO, GRPO, PSR and NSR across training steps, including (a) greedy decoding accuracy on the test set, (b) the model's entropy on the test set, (c) the ratio of correct responses per batch on the training set, and (d) the proportion of fully-solved prompts per batch (*i.e.*, all rollouts are correct) on the training set. NSR achieves competitive performance in greedy decoding accuracy while maintaining substantially higher entropy throughout training, suggesting greater exploration. NSR improves both the correct sample ratio and the fully-solved sample ratio, though less aggressively compared to other algorithms.

**RL training hurts Llama models' inference scaling performance.** As shown in Figure 4, applying different RL algorithms to `Llama-3.1-8B-Instruct` leads to degraded inference-scaling performance compared to the base model. Although Pass@1 improves on MATH and AMC23, Pass@256 drops substantially after RL training. This contrast between Qwen and Llama models highlights that backbone model itself often determines whether RL can bring benefits, consistent with recent works [72, 53, 68]. Among all methods, however, NSR causes the least degradation to the base model's scaling behavior.

## 4 Understanding the Effectiveness of Negative Sample Reinforcement

To better understand the learning mechanisms of PSR and NSR, and to explain why NSR consistently demonstrates strong inference scaling performance, we analyze their training dynamics through both empirical observations and gradient analysis.

### 4.1 Entropy as a Lens on Inference Scaling Performance

Since diversity is crucial for strong Pass@$k$ performance—especially at a large $k$—we quantify model diversity by tracking its entropy on a held-out test set throughout training, aiming to understand how entropy evolves under different training algorithms. In addition, we monitor two complementary metrics on the training set: the correct sample ratio, which measures the proportion of correct responses per batch, and the fully solved ratio, which measures the fraction of problems per batch where all rollouts are correct.

Figure 5b shows that NSR maintains a high level of entropy on the held out test set throughout training, closely matching that of the base model, while PSR leads to a rapid and substantial drop in entropy. PPO and GRPO offer a middle ground: they gradually reduce entropy during training, with levels that fall between those of PSR and NSR. Nevertheless, their entropy still declines considerably, which likely limits output diversity and helps explain why their Pass@256 performance remains below that of the base model. Interestingly, PPO exhibits a slight rebound in entropy during the later

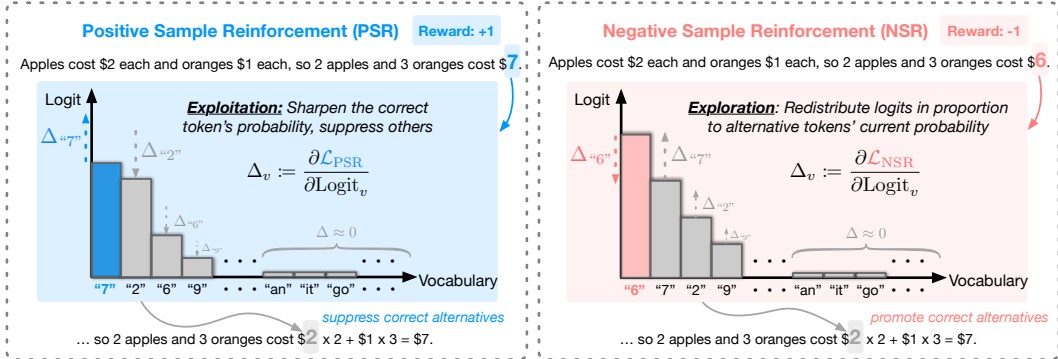

Figure 6: Gradient dynamics of PSR and NSR under a math word problem example. Bars indicate token logits before the update, and arrows indicate gradient direction and magnitude. Left (PSR): The model generates the correct response ("7") and receives a $+1$ reward. Gradients increase the logit of the sampled token while suppressing all others, including potentially correct alternatives like "2", resulting in a sharpened, overconfident distribution. Right (NSR): The model generates an incorrect response ("6") and receives a $-1$ reward. Gradients demote the sampled incorrect token and proportionally reallocate logits to other tokens (*e.g.*, "7", "2") based on their current probabilities, thereby promoting exploration on alternative correct paths and preserving diversity.

stages of training—a trend not seen in GRPO. This divergence may stem from PPO's use of a critic model, which can provide potentially more fine-grained and exploratory advantage estimates.

As shown in Figures 5c and 5d, NSR consistently improves the correct sample ratio but maintains a lower proportion of correct samples and fully solved problems throughout training. This indicates that the model avoids becoming overconfident in the observed correct responses. By preserving uncertainty, NSR enables stronger scaling performance in Pass@$k$, as demonstrated in Figure 2. In contrast, PSR rapidly increases the number of correct samples in each batch and achieves a higher fully solved ratio compared to other methods, suggesting a tendency to overfit to the observed correct responses. While this leads to better Pass@1 performance, it significantly reduces output diversity and results in weaker Pass@$k$ performance as $k$ increases.

These trends underscore the importance of preserving output entropy during RL and highlight a key insight: reinforcing negative samples alone can improve accuracy without compromising the base model's generation diversity.

### 4.2 Token-Level Gradient Dynamics of PSR and NSR

To gain a deeper understanding of the training dynamics of PSR and NSR, we conduct a token-level gradient analysis. The loss for both PSR and NSR on a training instance $(\boldsymbol{x}, \boldsymbol{y})$ takes the form:

$$\mathcal{L}(\theta) = -R \cdot \frac{1}{T} \sum_{t=1}^{T} \pi_\theta(y_t | \boldsymbol{x}, \boldsymbol{y}_{<t}) = -R \cdot \frac{1}{T} \sum_{t=1}^{T} \frac{\exp(z_{y_t})}{\sum_{v' \in \mathcal{V}} \exp(z_{v'})}, \tag{6}$$

where $R = r(\boldsymbol{x}, \boldsymbol{y}) \in \{-1, +1\}$ denotes the reward assigned to the sampled trajectory, and $z_v$ denotes the logit corresponding to token $v$ in the vocabulary $\mathcal{V}$.

To analyze the effect of these objectives on the model's token distribution, we compute the gradient of the loss with respect to token-level logits at each step. Let $\pi_v = \pi_\theta(v | \boldsymbol{x}, \boldsymbol{y}_{<t})$ denote the probability of token $v$ at time step $t$, the gradient descent directions of PSR and NSR are shown in Equations (7) and (8) (the full derivation is provided in Appendix A), which are illustrated in Figure 6.

$$-\frac{\partial \mathcal{L}_{\text{PSR}}}{\partial z_v} \propto \begin{cases} \pi_v \cdot (1 - \pi_v) & \text{if } v = y_t \quad \text{(sampled token)} \\ -\pi_{y_t} \cdot \pi_v & \text{if } v \neq y_t \quad \text{(unsampled token)} \end{cases} \tag{7}$$

This formulation clearly shows how PSR increases the logits of tokens appearing in correct responses while decreasing the logits of all other tokens. As training progresses, it repeatedly amplifies the probability of observed correct sequences—pushing their likelihoods toward 1, while suppressing alternative generations. This continual sharpening of the output distribution can lead to reduced entropy and overfitting as shown in Figure 5b, especially in on-policy settings where the same

examples may be encountered frequently. Over time, the model's behavior may collapse onto a narrow set of responses, limiting its ability to explore or generalize beyond what has been reinforced.

$$-\frac{\partial \mathcal{L}_{\text{NSR}}}{\partial z_v} \propto \begin{cases} -\pi_v \cdot \boxed{(1 - \pi_v)} & \text{if } v = y_t \quad \text{(sampled token)} \\ \pi_{y_t} \cdot \boxed{\pi_v} & \text{if } v \neq y_t \quad \text{(unsampled token)} \end{cases} \tag{8}$$

In contrast, NSR works by penalizing the logits of tokens in incorrect responses and softly redistributing probability mass to other candidate tokens. Importantly, NSR increases other tokens' logits in proportion to their current likelihoods. This objective has several desirable properties:

---

**Key Insights into NSR via Gradient Analysis**

1. **Preserving high-confidence priors**: When the model assigns high probability to certain tokens (i.e., $\pi_{y_t} \to 1$) that appear in incorrect outputs (*e.g.*, common grammatical or linguistic constructions), the negative gradient from NSR is scaled by $(1 - \pi_{y_t})$, resulting in small updates. This allows NSR to penalize mistakes without erasing fundamental knowledge learned during pretraining.

2. **Prior-guided probability redistribution**: NSR performs a soft reranking of the output distribution by boosting unsampled tokens' logits $z_v$ in proportion to their current probabilities $\pi_v$. This allows the model to effectively explore and search for better candidates according to their prior beliefs.

3. **Implicit regularization against overfitting**: NSR updates only when the model generates incorrect responses. Once the model consistently avoids these mistakes, NSR naturally halts further updates on those samples. This stopping criterion prevents the model from overfitting or collapsing diversity in examples it has already mastered.

---

These analyses highlight an important strength of NSR: it preserves the model's prior knowledge over plausible tokens and stops updating once errors are corrected, effectively locking in successful experiences. This is a unique advantage of NSR, as we demonstrate in Appendix B via gradient analysis that simply applying an entropy bonus in the policy loss, despite promoting diversity, cannot preserve model's prior. We also include a comparison with the widely used unlikelihood training objective, a previous approach designed to suppress undesirable or incorrect generations [57, 26].

### 4.3 Extending Gradient Analysis to PPO and GRPO

Our earlier analysis was based on a simple REINFORCE-like objective [58]. We now analyze how it extends to PPO and GRPO. The losses for GRPO and PPO are as follows:

$$\mathcal{L}_{\text{PPO}}(\theta) = -\frac{1}{T} \sum_{t=1}^{T} \min \left( \frac{\pi_\theta(y_t|\boldsymbol{x}, \boldsymbol{y}_{<t})}{\pi_{\text{old}}(y_t|\boldsymbol{x}, \boldsymbol{y}_{<t})} A_t, \text{clip}\left( \frac{\pi_\theta(y_t|\boldsymbol{x}, \boldsymbol{y}_{<t})}{\pi_{\text{old}}(y_t|\boldsymbol{x}, \boldsymbol{y}_{<t})}, 1 - \epsilon, 1 + \epsilon \right) A_t \right),$$

$$\mathcal{L}_{\text{GRPO}}(\theta) = -\frac{1}{G} \sum_{i=1}^{G} \frac{1}{T} \sum_{t=1}^{T} \Bigg( \min \left( \frac{\pi_\theta(y_{i,t}|\boldsymbol{x}, \boldsymbol{y}_{i,<t})}{\pi_{\text{old}}(y_{i,t}|\boldsymbol{x}, \boldsymbol{y}_{i,<t})} A_{i,t}, \text{clip}\left( \frac{\pi_\theta(y_{i,t}|\boldsymbol{x}, \boldsymbol{y}_{i,<t})}{\pi_{\text{old}}(y_{i,t}|\boldsymbol{x}, \boldsymbol{y}_{i,<t})}, 1 - \epsilon, 1 + \epsilon \right) A_{i,t} \right)$$
$$- \beta \text{KL}(\pi_\theta \| \pi_{\text{ref}}) \Bigg),$$

where $A$ denotes the advantage. Comparing the PPO and GRPO objectives to Equation (6), there are three key differences: (1) policy loss clipping, (2) KL regularization (reward penalty in PPO, loss in GRPO), and (3) advantages instead of raw rewards. We analyze their impact on the gradient dynamics below:

1. Clipping constrains update magnitude when the new policy diverges significantly from the old one, but preserves the gradient update direction, thus leaving our analysis qualitatively unchanged.

2. KL regularization discourages deviation from the reference policy. While it may dampen positive and negative updates, its practical impact is often negligible: for reasoning tasks, the KL coefficient is either very small or completely removed, leading to better performance, as demonstrated in recent works [6, 62, 69].

3. The advantage of GRPO $A_i = \frac{r_i - \text{mean}(\boldsymbol{r})}{\text{std}(\boldsymbol{r})}$ acts as a rescaling of the gradient and retains the raw reward's sign. In PPO, the advantage is computed relative to a value function: tokens that

Table 1: Pass@$k$ results on MATH, AIME 2025 and AMC23 with `Qwen2.5-Math-7B`. **Bold** and underlined numbers denote the best and second-best results for each $k$.

| Method | | | | | Pass@$k$ | | | | |
|---|---|---|---|---|---|---|---|---|---|
| $k$ | 1 | 2 | 4 | 8 | 16 | 32 | 64 | 128 | 256 |
| | | | | | **MATH** | | | | |
| Base Model | 63.2 | 76.0 | 83.7 | 88.4 | 91.6 | 93.7 | 95.2 | **96.2** | **96.9** |
| PPO | **76.6** | 82.6 | 86.7 | 89.6 | 91.7 | 93.4 | 94.7 | 95.6 | 96.3 |
| GRPO | 76.3 | 81.7 | 85.6 | 88.4 | 90.6 | 92.3 | 93.6 | 94.7 | 95.5 |
| REINFORCE | 74.8 | 79.1 | 82.4 | 84.9 | 86.9 | 88.5 | 89.9 | 91.1 | 92.0 |
| PSR | 74.1 | 78.3 | 81.6 | 84.1 | 86.2 | 87.9 | 89.3 | 90.4 | 91.2 |
| NSR | 75.7 | 82.4 | 86.9 | 90.1 | **92.4** | **94.1** | **95.3** | **96.2** | **96.9** |
| W-REINFORCE | **76.6** | **82.8** | **87.1** | **90.2** | **92.4** | **94.1** | **95.3** | 96.1 | 96.7 |
| | | | | | **AIME 2025** | | | | |
| Base Model | 6.1 | 9.7 | 13.8 | 17.9 | 22.2 | 26.5 | 30.8 | 36.6 | 46.7 |
| PPO | 8.5 | 13.2 | 18.0 | 22.5 | 26.6 | 30.3 | 33.8 | 37.9 | 43.3 |
| GRPO | 10.3 | 14.7 | 19.4 | 24.0 | 28.4 | 32.8 | 37.3 | 42.5 | 50.0 |
| REINFORCE | 9.2 | 12.5 | 16.3 | 21.0 | 26.4 | 32.3 | 38.6 | 44.7 | 50.0 |
| PSR | **11.6** | 14.1 | 16.2 | 18.6 | 21.7 | 25.7 | 30.9 | 36.9 | 43.3 |
| NSR | 10.0 | 14.6 | 19.2 | 24.1 | 29.3 | 34.6 | 40.2 | 46.0 | 53.3 |
| W-REINFORCE | 10.6 | **15.3** | **20.0** | **24.7** | **29.7** | **34.6** | **40.5** | **47.8** | **56.7** |
| | | | | | **AMC23** | | | | |
| Base Model | 41.0 | 56.2 | 69.2 | 78.9 | 85.1 | 89.1 | 92.9 | 97.2 | **100.0** |
| PPO | 62.0 | **70.0** | 76.1 | 80.9 | 85.3 | 89.5 | 93.1 | 96.0 | 97.5 |
| GRPO | 61.7 | 68.7 | 74.6 | 80.0 | 85.1 | 89.7 | 93.4 | 95.9 | 97.5 |
| REINFORCE | 61.8 | 68.5 | 73.4 | 77.0 | 80.8 | 85.1 | 89.3 | 91.9 | 92.5 |
| PSR | **62.6** | 69.9 | 74.5 | 77.5 | 80.3 | 83.5 | 87.2 | 90.6 | 92.5 |
| NSR | 60.9 | **70.0** | **77.4** | **83.2** | 87.6 | 91.1 | 94.5 | **97.9** | **100.0** |
| W-REINFORCE | 62.0 | **70.0** | 77.0 | 83.1 | **87.8** | **91.8** | **95.2** | 97.1 | 97.5 |

outperform the baseline are reinforced, while other tokens are penalized. This yields finer-grained credit assignment but does not alter the overall gradient dynamics—positive reinforcement still reduces diversity, and negative reinforcement promotes alternatives under the model prior.

Therefore, while PPO and GRPO introduce modifications that stabilize learning, the core gradient behaviors identified in our previous analysis still hold. We further discuss the differences between RLVR and RL with reward models in Appendix C.

## 5 Balancing Positive and Negative Reinforcement

Our previous analyses reveal a trade-off in reinforcement learning objectives: PSR improves Pass@1 quickly but sacrifices performance on Pass@$k$ for larger $k$, whereas NSR preserves Pass@$k$, but may underperform when $k$ is small. To strike a better balance between accuracy and diversity, we propose a simple weighted combination of PSR and NSR: we scale down the reward magnitude for PSR by a factor of $\lambda$ and combine it with NSR, allowing the model to learn from both correct and incorrect samples. When $\lambda = 1$, it is exactly the same as REINFORCE. We refer to this method as **Weighted-REINFORCE (W-REINFORCE)**:

$$\mathcal{L}_{\text{W-REINFORCE}}(\theta) = \underbrace{-\mathbb{E}_{\boldsymbol{x}\sim\mathcal{D}}\left[\sum_{\boldsymbol{y}:r(\boldsymbol{x},\boldsymbol{y})=1} \lambda \cdot \pi_\theta(\boldsymbol{y}|\boldsymbol{x})\right]}_{\lambda \cdot \mathcal{L}_{\text{PSR}}(\theta)} \underbrace{-\mathbb{E}_{\boldsymbol{x}\sim\mathcal{D}}\left[\sum_{\boldsymbol{y}:r(\boldsymbol{x},\boldsymbol{y})=-1} -\pi_\theta(\boldsymbol{y}|\boldsymbol{x})\right]}_{\mathcal{L}_{\text{NSR}}(\theta)}. \tag{9}$$

We apply $\lambda = 0.1$ in our experiments. As shown in Table 1, W-REINFORCE consistently delivers a favorable trade-off across a range of $k$ values on MATH, AIME 2025 and AMC23. On MATH, W-REINFORCE matches the best Pass@1 score 76.6 with PPO and has the highest performance for all $k \leq 64$, while still preserves competitive performance at $k = 256$. On AIME 2025, W-REINFORCE performs even more strongly—achieving the best result across all $k$ values except $k = 1$. These results confirm that W-REINFORCE, a simple weighted extension of the classic REINFORCE algorithm, strikes an effective balance between the strengths of PSR and NSR. By

merely scaling down the weight of positive rewards, it achieves strong performance while preserving diversity. Despite its simplicity, W-REINFORCE consistently outperforms strong RL algorithms such as PPO and GRPO across most Pass@$k$, while the vanilla REINFORCE underperforms. These findings suggest that this simple variant of REINFORCE can serve as a competitive alternative to more complex RL algorithms when the model prior is strong (*e.g.*, Qwen models). We provide an ablation study on the choice of $\lambda$ in Appendix E.

## 6 Related Work

**Reinforcement learning with verifiable rewards.** Reinforcement learning has shown great promise in improving large language models, as demonstrated by the success of reinforcement learning from human feedback (RLHF) and from AI feedback (RLAIF), which align model responses with human preferences [25, 32, 36, 39]. More recently, reinforcement learning with verifiable rewards [8, 14, 16, 23, 31, 43, 48, 52, 63, 69, 70, 71, 76, 54] has attracted growing attention for its effectiveness in incentivizing reasoning in LLMs with rule-based rewards [12, 15, 22, 47, 74, 10, 24, 34]. Notably, DeepSeek-R1 [15] and Kimi K1.5 [47] demonstrate that RLVR can elicit emergent reasoning behaviors such as long chain of thought and self-reflection, and achieve strong performance across diverse reasoning tasks such as math and coding problems. Yeo et al. [68] further explore the emergence of long CoT across different RLVR setups.

Despite these advances, many prior works mainly focus on evaluating the model's Pass@$1$ or greedy decoding performance, which might overlook the underlying change in model behavior (*e.g.*, inference scaling performance). More importantly, the mechanisms behind RLVR for driving reasoning and generalization remain underexplored. In this work, we take a step further by decomposing the RLVR objective into two distinct learning paradigms—learning from correct responses and learning from incorrect responses. We analyze the learning signal at the token level through gradient analysis and evaluate the models extensively using Pass@$k$ with a wide spectrum of $k$. Our findings highlight the critical yet previously underappreciated role of negative rewards in RLVR.

**Inference-time scaling behaviors.** Inference-time scaling has emerged as a promising direction for enhancing model performance [2, 3, 4, 29, 35, 38, 42, 46, 49, 50, 51, 56, 59, 80, 67, 73, 78, 55]— particularly in reasoning tasks where generating multiple candidate solutions [7, 80] or longer reasoning traces [35, 68, 75] can help hit the correct answer. Through the lens of inference-time scaling, recent studies have raised questions about whether RL-trained models are better than the base models [72, 18]. A recent work by [72] suggests that RLVR primarily adjusts the model's output distribution toward high-reward responses, rather than eliciting new reasoning abilities beyond the base model. By adopting Pass@$k$ as metric, they find that RL-trained models have inferior inference-time scaling performance than the base models. These findings align with our findings that reinforcing correct samples hurts Pass@$k$ at larger $k$. Another concurrent work by [9] studies supervised fine-tuning (SFT), and finds that diversity collapse during SFT adversely affects inference-time scaling performance. They show that SFT reduces generation diversity, leading to degraded Pass@$k$ performance, and that interpolating weights from early, potentially less overconfident checkpoints and later ones can effectively restore diversity and improve both Pass@$1$ and Pass@$k$.

In this work, we highlight the underestimated role of negative reinforcement in preserving and improving inference-time scaling performance, and that the accuracy and diversity trade-off in RLVR can be effectively balanced by adjusting positive and negative reward weights.

## 7 Conclusion

In this work, we investigate the mechanism underlying RLVR for LM reasoning. By decomposing RLVR into positive and negative sample reinforcement, we reveal a surprising finding: solely penalizing incorrect samples can effectively enhance LM reasoning capabilities while preserving generation diversity. Experimental results show that NSR consistently improves performance across a wide Pass@$k$ spectrum and in many cases matches or outperforms strong RL algorithms such as PPO and GRPO. Our gradient analysis demonstrates that NSR works by suppressing incorrect responses and redistributing probability mass toward plausible alternatives based on the model prior. Building on these findings, we proposed a simple variant of REINFORCE, Weighted-REINFORCE, that upweights the negative sample reinforcement. Empirical results show that it achieves a good balance between PSR and NSR, and yields consistent Pass@$k$ improvements across multiple reasoning benchmarks. We discuss limitations and future work directions in Appendix F.

## Acknowledgments

We would like to thank the members of the Princeton NLP Group for their valuable feedback and discussions. We also thank the anonymous reviewers and the area chair for their time, effort, and constructive suggestions, which have greatly improved the quality of this paper. This research is partially funded by the National Science Foundation (IIS-2211779), the NVIDIA Academic Grant, the Lambda Research Grant, and the OpenAI Superalignment Fast Grant. MX is supported by the Apple Scholars in AI/ML PhD Fellowship.

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

# A    Gradient Derivation

**Gradient of Equation (6).**    For simplicity of notation, let $\pi_v \coloneqq \pi_\theta(v|\boldsymbol{x}, y_{<t})$, and we omit the constant $\frac{1}{T}$ in the equation which effectively scales the gradient magnitude.

$$\frac{\partial \mathcal{L}}{\partial z_v} = -R \cdot \frac{\mathbb{1}(v = y_t) \exp(z_{y_t}) \sum_{v' \in \mathcal{V}} \exp(z_v) - \exp(z_{y_t}) \exp(z_v)}{\left(\sum_{v' \in \mathcal{V}} \exp(z_{v'})\right)^2}$$

For the sampled token $v = y_t$, the gradient simplifies to

$$\begin{aligned}
\frac{\partial \mathcal{L}}{\partial z_v} &= -R \cdot \frac{\exp(z_v) \sum_{v' \in \mathcal{V}} \exp(z_{v'}) - \exp(z_v)^2}{\left(\sum_{v' \in \mathcal{V}} \exp(z_{v'})\right)^2} \\
&= -R \cdot \frac{\exp(z_v) \left(\sum_{v' \in \mathcal{V}} \exp(z_{v'}) - \exp(z_v)\right)}{\left(\sum_{v' \in \mathcal{V}} \exp(z_{v'})\right)^2} \\
&= -R \cdot \left(\frac{\exp(z_v)}{\sum_{v' \in \mathcal{V}} \exp(z_{v'})} \cdot \frac{\sum_{v' \in \mathcal{V}} \exp(z_{v'}) - \exp(z_v)}{\sum_{v' \in \mathcal{V}} \exp(z_{v'})}\right) \\
&= -R \cdot \pi_v \cdot (1 - \pi_v)
\end{aligned}$$

For the unsampled token $v \neq y_t$, the gradient simplifies to

$$\begin{aligned}
\frac{\partial \mathcal{L}}{\partial z_v} &= -R \cdot \frac{-\exp(z_{y_t}) \exp(z_v)}{\left(\sum_{v' \in \mathcal{V}} \exp(z_{v'})\right)^2} \\
&= R \cdot \frac{\exp(z_{y_t})}{\sum_{v' \in \mathcal{V}} \exp(z_{v'})} \cdot \frac{\exp(z_v)}{\sum_{v' \in \mathcal{V}} \exp(z_{v'})} \\
&= R \cdot \pi_{y_t} \cdot \pi_v
\end{aligned}$$

In summary, the gradient of Equation (6) can be expressed as

$$\frac{\partial \mathcal{L}}{\partial z_v} = \begin{cases} -R \cdot \pi_v \cdot (1 - \pi_v) & \text{if } v = y_t \quad \text{(sampled token)} \\ R \cdot \pi_{y_t} \cdot \pi_v & \text{if } v \neq y_t \quad \text{(unsampled token)} \end{cases}$$

# B    NSR differs from Entropy and Unlikelihood Training in Gradients

**Gradient of entropy regularization.**    Entropy regularization is also one way to induce a diverse output probability distribution. We compute and analyze its gradient dynamics to reveal its difference from the NSR objective as follows. The entropy loss is

$$\mathcal{H}(\pi_\theta) = -\sum_{v' \in \mathcal{V}} \pi_{v'} \log \pi_{v'},$$

and the gradient ascent direction to maximize entropy is

$$\frac{\partial \mathcal{H}}{\partial z_v} = -\sum_{v' \in \mathcal{V}} \frac{\partial \pi_{v'}}{\partial z_v}(1 + \log \pi_{v'})$$

$$= -\sum_{v' \in \mathcal{V}} \pi_{v'}\left(\mathbb{1}(v = v') - \pi_v\right)(1 + \log \pi_{v'})$$

$$= -\left(\pi_v(1 + \log \pi_v) - \pi_v \sum_{v' \in \mathcal{V}} \pi_{v'}(1 + \log \pi_{v'})\right)$$

$$= -\pi_v \left(\log \pi_v - \underbrace{\sum_{v' \in \mathcal{V}} \pi_{v'} \log \pi_{v'}}_{=\bar{\ell}}\right)$$

It can be seen that the gradient sign and magnitude depend on how the log-probability of token $v$, $\log \pi_v$, compares to the expected log probability $\bar{\ell}$ over the vocabulary: when token $v$ is significantly more probable than expected, it receives a larger negative gradient, pushing its logit down more strongly. In contrast, when token $v$ is less probable than average, it receives a positive gradient that boosts its logit.

As a result, entropy maximization systematically flattens the output distribution by suppressing high-confidence tokens and boosting low-confidence ones. This behavior can conflict with the model's prior knowledge: confidently correct tokens—such as syntactically or semantically appropriate completions—are penalized more for being too probable, while completely implausible tokens may be promoted simply because they are unlikely under the current policy.

**Gradient of unlikelihood training objective.** While NSR shares the goal of penalizing undesirable outputs with unlikelihood training [57, 26], their formulation and resulting dynamics are fundamentally different: NSR penalizes incorrect outputs using their raw probabilities, rather than minimizing $-\log(1 - \pi_\theta(\boldsymbol{y}|\boldsymbol{x}))$ as in unlikelihood training. This distinction results in different gradient behavior and training dynamics as shown below:

$$\mathcal{L}_{\text{unlikelihood}}(\theta) = -\frac{1}{T}\sum_{t=1}^{T} \log\left(1 - \pi_\theta(y_t \mid \boldsymbol{x}, \boldsymbol{y}_{<t})\right). \tag{10}$$

Taking the derivative w.r.t. token logits $z_v$ yields:

$$-\frac{\partial \mathcal{L}_{\text{unlikelihood}}}{\partial z_v} \propto \begin{cases} -\pi_v & \text{if } v = y_t \quad \text{(sampled token)} \\ \dfrac{\pi_{y_t}}{1 - \pi_{y_t}} \cdot \pi_v & \text{if } v \neq y_t \quad \text{(unsampled token)} \end{cases} \tag{11}$$

Comparing Equation (11) against Equation (8), it can be seen that Equation (11) induces a markedly different update: it pushes down the sampled token $y_t$'s logit and redistributes probability mass to other tokens in proportion to their current probabilities $\pi_v$. Consequently, confident predictions are penalized more aggressively, which may erode prior knowledge. In contrast, the gradient of NSR (Equation (8)) is dampened by a $(1 - \pi_v)$ factor. This has the opposite effect: as confidence in a sampled token grows, the gradient update shrinks when that token appears in an incorrect answer. By reducing the penalty on high-confidence tokens, NSR avoids destructive updates and effectively preserves pretrained knowledge.

## C   RLVR vs. RL with Reward Models

Different from RLVR, many RL setups require training reward models to assign scalar values that indicate the quality of generation (*e.g.*, in RLHF [36]). During policy optimization, these rewards are often normalized within a batch by subtracting the mean, so that a sample's final reward depends on its relative ranking compared to others in the same batch. As a result, a sample's reward can change sign (from positive to negative or vice versa) depending on the overall quality of the batch, making it difficult to interpret rewards as absolute indicators of positive or negative samples. In contrast, RLVR

applies a binary reward $(+1/-1)$. The signs of the rewards are inherently tied to the correctness of individual sequences, as each token in a sequence receives the same reward and the batch average reward always remains within $[-1, 1]$, thus reward normalization will preserve the original sign. This inherent sign preservation in RLVR is critical for our analysis, as it enables a clean and unambiguous separation into positive and negative learning paradigms based on the reward signs.

## D  Implementation Details

### D.1  Objectives and Training Hyperparameters

The objectives of PPO, GRPO, PSR and NSR are as follows:

$$\mathcal{L}_{\text{PPO}}(\theta) = -\frac{1}{T}\sum_{t=1}^{T}\min\left(\frac{\pi_\theta(y_t|\boldsymbol{x}, \boldsymbol{y}_{<t})}{\pi_{\text{old}}(y_t|\boldsymbol{x}, \boldsymbol{y}_{<t})}A_t, \text{clip}\left(\frac{\pi_\theta(y_t|\boldsymbol{x}, \boldsymbol{y}_{<t})}{\pi_{\text{old}}(y_t|\boldsymbol{x}, \boldsymbol{y}_{<t})}, 1-\epsilon, 1+\epsilon\right)A_t\right),$$

$$\mathcal{L}_{\text{GRPO}}(\theta) = -\frac{1}{G}\sum_{i=1}^{G}\frac{1}{T}\sum_{t=1}^{T}\left(\min\left(\frac{\pi_\theta(y_{i,t}|\boldsymbol{x}, \boldsymbol{y}_{i,<t})}{\pi_{\text{old}}(y_{i,t}|\boldsymbol{x}, \boldsymbol{y}_{i,<t})}A_{i,t}, \text{clip}\left(\frac{\pi_\theta(y_{i,t}|\boldsymbol{x}, \boldsymbol{y}_{i,<t})}{\pi_{\text{old}}(y_{i,t}|\boldsymbol{x}, \boldsymbol{y}_{i,<t})}, 1-\epsilon, 1+\epsilon\right)A_{i,t}\right)\right.$$
$$\left. - \beta\text{KL}(\pi_\theta\|\pi_{\text{ref}})\right),$$

$$\mathcal{L}_{\text{PSR}}(\theta) = -\frac{1}{T}\sum_{t=1}^{T}\min\left(\frac{\pi_\theta(y_t|\boldsymbol{x}, \boldsymbol{y}_{<t})}{\pi_{\text{old}}(y_t|\boldsymbol{x}, \boldsymbol{y}_{<t})}, \text{clip}\left(\frac{\pi_\theta(y_t|\boldsymbol{x}, \boldsymbol{y}_{<t})}{\pi_{\text{old}}(y_t|\boldsymbol{x}, \boldsymbol{y}_{<t})}, 1-\epsilon, 1+\epsilon\right)\right), \quad r(\boldsymbol{x}, \boldsymbol{y}) = 1,$$

$$\mathcal{L}_{\text{NSR}}(\theta) = -\frac{1}{T}\sum_{t=1}^{T}\min\left(-\frac{\pi_\theta(y_t|\boldsymbol{x}, \boldsymbol{y}_{<t})}{\pi_{\text{old}}(y_t|\boldsymbol{x}, \boldsymbol{y}_{<t})}, -\text{clip}\left(\frac{\pi_\theta(y_t|\boldsymbol{x}, \boldsymbol{y}_{<t})}{\pi_{\text{old}}(y_t|\boldsymbol{x}, \boldsymbol{y}_{<t})}, 1-\epsilon, 1+\epsilon\right)\right), \quad r(\boldsymbol{x}, \boldsymbol{y}) = -1,$$

where $\epsilon$ is the clip ratio, and $\beta$ is the KL penalty coefficient. For PPO, the KL penalty is applied to the advantage. For GRPO, the KL penalty is added to the final loss. Both PPO and GRPO use a KL penalty coefficient of 1e-3. For PSR and NSR, we do not apply KL penalty, which we find to result in better performance. The learning rate of the critic model in PPO is 1e-5. The clip ratio is set to 0.2. We also apply entropy bonus to all the above objectives, with a coefficient of 1e-4. Our experiments are conducted over a single node with 8 NVIDIA H200 GPUs.

### D.2  Prompt Templates

We adopt the prompt templates for Qwen models following [74], as shown in Tables 2 and 3.

Table 2: Prompt template for `Qwen2.5-Math-7B`.

```
<|im_start|>system
You are a helpful assistant.<|im_end|>
<|im_start|>user
{input}
Please reason step by step, and put your final answer within \boxed{}.<|im_end|>
<|im_start|>assistant
```

### D.3  Advantage Normalization

The implementation of computing advantage in verl includes an advantage normalization. However, for PSR and NSR, since we exclude the KL penalty, the advantage is equal to the raw reward (*i.e.*, $+1$ for PSR and $-1$ for NSR). In this case, applying normalization would cause the advantage to be zero, thus eliminating the learning signal. As a result, we disable this normalization in PSR and NSR's implementation. A recent work [61] also suggests that the reward normalization may not be necessary for certain settings.

Table 3: Prompt template for `Qwen3-4B` non-thinking mode.

```
<|im_start|>system
You are a helpful assistant.<|im_end|>
<|im_start|>user
{input}
Please reason step by step, and put your final answer within \boxed{}.<|im_end|>
<|im_start|>assistant
<think>

</think>
```

Table 4: Pass@$k$ results on MATH, AIME 2025 and AMC23 with different $\lambda$ using `Qwen2.5-Math-7B`. **Bold** and underlined numbers denote the best and second-best results for each $k$.

| | | | | | Pass@$k$ | | | | |
|---|---|---|---|---|---|---|---|---|---|
| | 1 | 2 | 4 | 8 | 16 | 32 | 64 | 128 | 256 |
| $\lambda$ | | | | | **MATH** | | | | |
| 0 | 75.7 | 82.4 | 86.9 | 90.1 | **92.4** | **94.1** | **95.3** | **96.2** | **96.9** |
| 0.05 | 75.6 | 82.2 | 86.7 | 89.9 | 92.2 | 94.0 | 95.3 | 96.3 | 97.1 |
| 0.1 | **76.6** | **82.8** | **87.1** | **90.2** | **92.4** | **94.1** | **95.3** | 96.1 | 96.7 |
| 0.2 | 75.8 | 81.2 | 85.2 | 88.3 | 90.7 | 92.6 | 94.0 | 95.1 | 95.9 |
| 1 | 74.8 | 79.1 | 82.4 | 84.9 | 86.9 | 88.5 | 89.9 | 91.1 | 92.0 |
| $\lambda$ | | | | | **AIME 2025** | | | | |
| 0 | 10.0 | 14.6 | 19.2 | 24.1 | 29.3 | 34.6 | 40.2 | 46.0 | 53.3 |
| 0.05 | 9.0 | 13.6 | 18.3 | 22.9 | 27.8 | 33.3 | 39.4 | 46.2 | 53.3 |
| 0.1 | 10.6 | **15.3** | **20.0** | **24.7** | **29.7** | **34.6** | **40.5** | **47.8** | **56.7** |
| 0.2 | 9.4 | 13.2 | 17.3 | 21.8 | 26.5 | 31.1 | 35.7 | 39.6 | 43.3 |
| 1 | 9.2 | 12.5 | 16.3 | 21.0 | 26.4 | 32.3 | 38.6 | 44.7 | 50.0 |
| $\lambda$ | | | | | **AMC23** | | | | |
| 0 | 60.9 | **70.0** | **77.4** | **83.2** | 87.6 | 91.1 | 94.5 | **97.9** | **100.0** |
| 0.05 | 61.6 | 70.9 | 77.8 | 82.9 | 87.1 | 91.2 | 95.1 | 98.1 | 100.0 |
| 0.1 | 62.0 | **70.0** | 77.0 | 83.1 | **87.8** | **91.8** | **95.2** | 97.1 | 97.5 |
| 0.2 | 60.4 | 68.0 | 73.9 | 79.3 | 84.5 | 89.5 | 93.6 | 96.4 | 97.5 |
| 1 | 61.8 | 68.5 | 73.4 | 77.0 | 80.8 | 85.1 | 89.3 | 91.9 | 92.5 |

# E   Impact of Reward Weighting Coefficient $\lambda$

To examine how sensitive W-REINFORCE is to the choice of $\lambda$, we conduct an ablation study with $\lambda \in {0, 0.05, 0.1, 0.2, 1}$ on the MATH dataset, with $\lambda = 0$ corresponding to NSR and $\lambda = 1$ corresponding to vanilla REINFORCE. As shown in Table 4, performance is relatively stable when $\lambda \leq 0.2$, while larger values gradually degrade Pass@$k$ across all $k$. This confirms that the key to W-REINFORCE 's effectiveness lies in moderately down-weighting the positive reward, which prevents the model from collapsing its distribution toward the highest-probability answers. When $\lambda = 1$, Pass@256 drops substantially, underscoring that excessive emphasis on positive rewards harms exploration and diversity. Overall, these results suggest that W-REINFORCE is robust to the choice of $\lambda$, and a small $\lambda$ can offer a better trade-off between Pass@1 and Pass@256.

# F   Limitations

We discuss the limitations of our work and future work directions as follows.

1. **Performance degradation with extended NSR training.** We observe that extensive training with NSR (*e.g.*, over hundreds of gradient steps) leads to a noticeable decline in performance, whereas W-REINFORCE does not exhibit this issue. However, such instability is not unique to NSR—standard RL algorithms like GRPO also experience similar degradation or training

collapse under extended updates, as studied in recent works [30, 19, 79]. This suggests that NSR's implicit mechanism for preserving prior knowledge may be insufficient to ensure stable training over the long term, and incorporating some degree of PSR may be necessary. Future work could explore more sophisticated methods for integrating NSR and PSR to design more robust RL algorithms, and investigate NSR as a potential warm-up phase before standard RL training given that it improves the full Pass@$k$ spectrum.

2. **Beyond sparse and binary rewards.** In this work, we focus on the RLVR setup with sparse and binary outcome rewards. However, many real-world tasks require dense reward signals that offer more fine-grained feedback (*e.g.*, evaluating intermediate reasoning steps or subjective tasks). It remains an open question how PSR, NSR, or W-REINFORCE would perform in such settings, where the learning signals are continuous rather than discrete and potentially more difficult to interpret.

