# OpenReview forum: "The Surprising Effectiveness of Negative Reinforcement in LLM Reasoning"
_NeurIPS.cc/2025/Conference — NeurIPS 2025 poster_

### Official Review · Reviewer_nxZP · 2025-06-23

**Clarity:** 3
**Significance:** 2
**Originality:** 2
**Rating:** 4
**Confidence:** 4

**Summary:**

This paper delves into the effectiveness of negative reinforcement in enhancing the reasoning capabilities of LLMs. It dissects RLVR into PSR and NSR, exploring their distinct impacts on model behavior and generalization. The authors demonstrate that NSR, which solely penalizes incorrect samples, can surprisingly elevate model performance across the Pass@k spectrum, often matching or surpassing traditional RL algorithms like PPO and GRPO. Based on these insights, the paper proposes W-REINFORCE, a modified RL objective that emphasizes NSR, achieving notable performance improvements across multiple reasoning benchmarks.

**Questions:**

1. Can the authors elaborate on the relationship between SFT and PSR?
2. How does the proposed W-REINFORCE algorithm compare to other RL variants in the literature that also emphasize negative reinforcement? What specific advantages does it offer over these existing methods?
3. The choice of the weighting factor $\lambda$  in W-REINFORCE seems crucial for its performance. Could the authors offer more detailed guidance or a systematic approach for selecting an appropriate $\lambda$ value, especially for different models and tasks?
4. The authors mention that NSR refines the model's existing knowledge rather than introducing entirely new behaviors. Could they provide more in-depth analysis or visualization to help readers better understand this refinement process?
5. Why does W-REINFORCE underperform PPO in pass@1? An explanation from the authors would be appreciated.
6. How does the proposed method generalize to OOD datasets? *(This will not influence my score)*
7. What is the specific value of $n$ in the pass@k evaluation metric? (Is it 256, as suggested in line 111?)
If $k=256$, based on the definition in Eq. (5), the pass@256 values in Table 1 should be binary (either 0 or 1). This is because if we assume $n=k=256$, the term $\binom{n}{k}$ becomes $\binom{256}{256} = 1$. Consequently, the term $\binom{n-c}{k}$ becomes $\binom{256-c}{256}$, which evaluates to 0 for any $c > 0$ and to 1 only when $c=0$. This seems to contradict the results shown in Table 1.

**Ethical Concerns:**

["NO or VERY MINOR ethics concerns only"]

**Final Justification:**

The authors' responses addressed most of my concerns.

**Limitations:**

Yes.

**Paper Formatting Concerns:**

None.

**Quality:**

2

**Strengths And Weaknesses:**

**Strengths:**
- The paper presents a novel perspective by decomposing RLVR into PSR and NSR, offering valuable insights into their individual roles in shaping model behavior.
- The authors conduct extensive experiments to validate their claims, providing a comprehensive analysis of how PSR and NSR affect model performance across different metrics.
- The findings have important implications for the design of RL algorithms for LLMs, particularly in preserving diversity and promoting generalization.

**Weaknesses:**
- While the decomposition of RLVR into PSR and NSR is insightful, the overall concept of this method is not entirely new and has been touched upon in previous studies [1].
- The proposed W-REINFORCE algorithm, despite showing promise, requires careful tuning of the weighting factor $\lambda$ . The paper does not provide sufficient guidance on how to optimally set this parameter in practice.
- The claim that NSR (using only negative samples) outperforms PSR is a bit counterintuitive and confusing. The authors should provide a clearer explanation for this result (Table 1).

[1] Xiong, Wei, et al. "A minimalist approach to llm reasoning: from rejection sampling to reinforce." arXiv preprint arXiv:2504.11343 (2025).

---

> ### Author Rebuttal · Authors · 2025-07-31
>
> Thanks for your thoughtful feedback. We address your raised issues as follows.
>
> > **While the decomposition of RLVR into PSR and NSR is insightful, the overall concept of this method is not entirely new and has been touched upon in previous studies [1].**
>
> According to NeurIPS policies, works posted after March 1st, 2025 are considered concurrent. Reference [1] was posted on arXiv on April 15 and should therefore be treated as concurrent, not prior work.
>
> More importantly, the methodology in [1] is fundamentally different from ours. Their approach is based on **prompt filtering**: removing prompts whose responses are entirely correct or incorrect, this means that the training data still contains a **mix of both positive and negative reward examples**. In contrast, our work introduces a **complete objective decomposition** of RLVR into PSR and NSR, fully isolating positive and negative reward examples. This clean setup allows us to clearly examine how each reward type individually shapes model behavior.
>
> > **The proposed W-REINFORCE algorithm, despite showing promise, requires careful tuning of the weighting factor**
>
> > **The choice of the weighting factor in W-REINFORCE seems crucial for its performance. Could the authors offer more detailed guidance or a systematic approach for selecting an appropriate value, especially for different models and tasks?**
>
> We provide hyperparameter study of W-REINFORCE with different λ as follows:
>
> MATH (Pass@k)
>
> | λ   | 1        | 2        | 4        | 8        | 16       | 32       | 64       | 128      | 256      |
> | ---- | -------- | -------- | -------- | -------- | -------- | -------- | -------- | -------- | -------- |
> | 0.05 | 75.6     | 82.2     | 86.7     | 89.9     | 92.2     | 94.0     | 95.0     | **96.3** | **97.1** |
> | 0.1  | **76.6** | **82.8** | **87.1** | **90.2** | **92.4** | **94.1** | **95.3** | 96.1     | 96.7     |
> | 0.2  | 75.8     | 81.2     | 85.2     | 88.3     | 90.7     | 92.6     | 94.0     | 95.1     | 95.9     |
>
> RL algorithms generally require hyperparameter tuning, but our experiments show that W-REINFORCE is not highly sensitive to λ.
>
> > **The claim that NSR (using only negative samples) outperforms PSR is a bit counterintuitive and confusing. The authors should provide a clearer explanation for this result (Table 1).**
>
> The explanation why NSR can outperform PSR is described in Line 142 and further supported by empirical evidence in Figure 3 and analysis in Section 4.2. As shown in Figure 3, PSR quickly overfits to the observed correct rollouts, which reduces output diversity and harms exploration, resulting in inferior pass@k performance when k is large. In contrast, NSR penalizes incorrect outputs and maintains a high entropy, preserving a broader search space. Section 4.2 further explains this with a principled gradient analysis. It reveals that PSR suppresses all possible alternatives except the highest probability token, while NSR encourages exploration by promoting all plausible candidates based on model’s prior. These results and analysis explain why NSR can outperform PSR.
>
> > **Can the authors elaborate on the relationship between SFT and PSR?**
>
> SFT typically refers to off-policy training, whereas PSR is fully on-policy. Moreover, PSR follows a standard RL setting that incorporates importance sampling and clipping in its loss function.
>
> > **How does the proposed W-REINFORCE algorithm compare to other RL variants in the literature that also emphasize negative reinforcement? What specific advantages does it offer over these existing methods?**
>
> To the best of our knowledge, our work is the first to find that solely penalizing mistakes through negative reinforcement learning, without relying on positive reward signals, can effectively improve LLM reasoning. If we missed any relevant work, we are happy to discuss and cite them accordingly.
>
> The advantages are that it can preserve models’ generation diversity and strike a good balance between pass@1 and pass@k performance when k is large (i.e., more decoding budgets).
>
> > **The authors mention that NSR refines the model's existing knowledge rather than introducing entirely new behaviors. Could they provide more in-depth analysis or visualization to help readers better understand this refinement process?**
>
> Section 4.2 provides a principled gradient analysis on why NSR is refining existing knowledge. It reveals that the gradients of NSR promote all plausible alternatives based on models’ prior. Figure 4 offers a visualization of this process.
>
> > **Why does W-REINFORCE underperform PPO in pass@1? An explanation from the authors would be appreciated.**
>
> As shown in table 1, W-REINFORCE is either comparable to PPO (on MATH and AMC23) or outperforms PPO (on AIME 2025) measured by pass@1.
>
> > **How does the proposed method generalize to OOD datasets?**
>
> We train models on the MATH dataset and evaluate them on AIME 2025 and AMC 23. MATH consists of PhD-level problems, while AIME and AMC feature Olympiad-level math questions with different formats and difficulty distributions. These test sets can be considered OOD relative to the training data. The results on various benchmarks suggest our method can generalize to OOD datasets.
>
> > **What is the specific value of n in the pass@k evaluation metric?**
>
> Yes, n is 256. Pass@256 is computed by averaging over the entire evaluation set. For example, the MATH test set contains 5000 problems. For each problem, pass@256 is either 0 or 1 depending on whether at least one of the 256 generations is correct. The final pass@256 metric reported in Table 1 is the mean across all problems, which yields a number between 0 and 1.
>
> [1] Xiong, Wei, et al. "A minimalist approach to llm reasoning: from rejection sampling to reinforce." 2025.

---

> > ### Comment · Reviewer_nxZP · 2025-08-01
> >
> > Thanks for the reply, which addressed most of my concerns. I updated my score.
> >
> > One quick question on Section 4.2: your analysis explains why NSR encourages exploration, but it's not entirely clear why using **only** negative samples surpasses methods with positive ones. This is especially interesting since techniques like self-imitation learning [1] show clear benefits from reusing **positive samples** in online RL. Could you briefly discuss this difference? This won't affect my score.
> >
> > [1]  Oh, Junhyuk, et al. "Self-imitation learning." International conference on machine learning. PMLR, 2018.

---

> > > ### Author Response · Authors · 2025-08-04
> > >
> > > Thank you for updating the score and insightful question. The key difference lies in the strength of the model's prior. Modern LLMs already possess substantial prior knowledge through extensive pre-training and post-training, which means they may have the knowledge to solve the problem, therefore, simply penalizing incorrect outputs (as in NSR) is sufficient to guide the model toward correct behavior. In contrast, techniques like self-imitation learning [1] are typically applied in settings where the agent is trained from scratch with randomly initialized parameters and no prior knowledge. In such cases, leveraging positive trajectories is essential for learning because the agent lacks any inherent ability to generate useful outputs. Therefore, while positive sample reuse is crucial in classical RL from scratch, its impact on the RLVR setting for today’s LLMs necessitates a re-examination, and we demonstrate in this work that over-optimizing on known correct outputs can reduce diversity and hurt the performance of LLM reasoning.
> > >
> > > We will incorporate the discussions above in the revision. Thanks again for your time!
> > >
> > > [1] Oh, Junhyuk, et al. "Self-imitation learning." International conference on machine learning. PMLR, 2018.

---

### Official Review · Reviewer_hPq3 · 2025-06-24

**Clarity:** 3
**Significance:** 3
**Originality:** 3
**Rating:** 5
**Confidence:** 4

**Summary:**

Studying RL with verifiable (binary output-based) rewards, the authors analyze the impact of positive and negative reward signals on model performance and compare to standard methods such as PPO and GRPO. The authors focus the evaluation on math reasoning using the MATH, AIME2025, and AMC23 datasets.

The authors empirically show using Qwen2.5-Math-7B with a full spectrum of Pass@k with k in {1, 2, 4, 8…, 256} that only using positive reward samples during training leads to better performance for pass@1 while reducing diversity of responses and overfitting for larger values of k. The authors find that using negative reward samples performs similarly to PPO and GRPO methods in terms of larger k values and that it outperforms other methods in preserving model entropy (response diversity) by reducing the likelihood of wrong responses and reallocating the probability mass to others. The authors also use token-level gradient dynamics to verify these claims.

The authors also introduce a weighted variant of REINFORCE that prioritizes negative sample that contributes to the loss function with a linear hyperparameter, which performs comparably to more elaborate algorithms like GRPO and PPO on the tested benchmarks.

Based on these results, the authors claim that negative reward samples naturally avoid overfitting (unlike positive reward samples), allow the model to effectively explore and search according to base model prior beliefs, and that the accuracy and diversity trade-off in RLVR is a balance issue between positive and negative rewards (which can be solved by re-weighting).

**Questions:**

- What are the implications of splitting rewards into “only positive” and “only negative”? Besides the direct impact of positive and negative samples on training curves, how does this simplification neglect the indirect impact of having positive and negative samples during training? My question is motivated by the assumption that LMs in RLVR encounter more wrong than correct answers and, e.g., does encountering a positive sample every 10 negative samples impact the overall trajectory differently when comparing trajectories with only positive or negative samples?

- To what extent are the results in Figure 2 explained by PPO and GRPO training encountering more negative than positive reward samples during training?

- To what extent is the trend of RL models losing their advantage at larger decoding budgets solely explained by negative reward samples (as argued in lines 136-141)? Are there no other possible explanations?

- It is unclear to me if the authors agree with this statement “[line 265] RLVR primarily adjusts the model’s output distribution toward high-reward responses, rather than eliciting new reasoning abilities beyond the base model.” Based on the discussed results, I think this is ambiguous, as it matters not just that the probability mass is redistributed, but how.  Please clarify your position and argumentation.

- What can we learn or hypothesize from your study about RL algorithms using dense/progress-based rewards?

Minor comment:
- The x-axis labels are missing in Figure 3.

**Ethical Concerns:**

["NO or VERY MINOR ethics concerns only"]

**Final Justification:**

The authors successfully resolved my concerns/reasons to reject, in particular, by providing statistical significance tests, running new experiments, and resolving uncertainties I've had. My original reasons to accept remain unchanged and are strengthened by the provided author revisions and rebuttal.

**Limitations:**

The authors provide a very brief limitation section in Appendix C. *[For the AC]* I am not sure whether the limitation section is supposed to be in the main body of the paper or not.

The authors acknowledge the limitation of only using one model to build their results and that they neglect of dense/intermediate rewards.

**Paper Formatting Concerns:**

No formatting issues

**Quality:**

3

**Strengths And Weaknesses:**

Reasons to accept:

- This work potentially provides crucial insight into outcome-based rewards in RL for LLMs that can have a significant impact for similar methods that are limited to not using dense/progress rewards. Understanding the importance of negative samples can also have a significant impact on alignment and safety methodologies, where constructing negative samples based on a rule or reason is easier than constructing positively rewarded ones.

- Good methodology using entropy and token-level gradients (quality).

- The paper is well written and, to the best of my knowledge, original work.

Reason to reject:

- My main concerns are related to the lack of statistical significance tests or statistical uncertainty estimates (quality, uncertain significance). First, the authors claim to have fulfilled the NeurIPS requirement for statistical significance/uncertainty estimates with the reason ”Our results are based on 256 random samplings and calculated with an unbiased estimator.”. *Just using an unbiased estimator does not eliminate the requirement for uncertainty estimates/statistical significance tests!*

Doing a few estimates, (assuming k=10, n=256, for a few c values), I find that the statistical uncertainty could be in the order of magnitude of 3-8 percent, i.e., in the order of magnitude the gained improvements in Table 1 are claimed or for which the differences in Figures 2 and 3 are used as evidence. I could be wrong, though, so I would require the authors to demonstrate that their results are statistically significant using the method they deem most appropriate.

Without demonstrating that the empirical results used in Figures 2, 3, and Table 1 are statistically significant and not just statistical trends, I have to question the validity of the results of the paper. The statistical trends are definitely there, but it has not been shown that these trends are actually statistically significant (at least 2-sigma or 95% confidence level deviations). For example, all claims in section 3.2 should either be highlighted as just observed trends (not results) or require the derivation of an uncertainty estimator + reported uncertainties. Table 1 should also include the statistical uncertainties. This is my main concern, and fully addressing this point will make me increase my score to a 4 or 5 (together with the quality and significance score).

- The authors only tested one model (quality, uncertain significance). I realize that expecting multiple RL training runs for models above 7B in academic papers is unreasonable. However, I think calling for testing at least a second 7B or some of the smaller 3B models from different model families to support universal empirical RLxLM training dynamics claims is not.

---

> ### Author Rebuttal · Authors · 2025-07-31
>
> Thanks for your thoughtful feedback. We address your raised issues as follows.
>
> > **My main concerns are related to the lack of statistical significance tests or statistical uncertainty estimates (quality, uncertain significance)...**
>
> We report the 95% confidence intervals on MATH with Qwen2.5-Math-7B in the table below:
>
> | Method | 1 | 2 | 4 | 8 | 16 | 32 | 64 | 128 | 256 |
> | --- | --- | --- | --- | --- | --- | --- | --- | --- | --- |
> | GRPO | 76.3 ± 1.0 | 81.7 ± 0.9 | 85.6 ± 0.9 | 88.4 ± 0.8 | 90.6 ± 0.7 | 92.3 ± 0.7 | 93.6 ± 0.6 | 94.6 ± 0.6 | 95.5 ± 0.6 |
> | PPO | **76.6 ± 1.0** | 82.6 ± 0.9 | 86.7 ± 0.8 | 89.6 ± 0.7 | 91.7 ± 0.7 | 93.4 ± 0.6 | 94.7 ± 0.6 | 95.6 ± 0.5 | 96.3 ± 0.5 |
> | PSR | 74.1 ± 1.1 | 78.3 ± 1.0 | 81.6 ± 1.0 | 84.1 ± 0.9 | 86.2 ± 0.9 | 87.9 ± 0.8 | 89.3 ± 0.8 | 90.3 ± 0.8 | 91.2 ± 0.8 |
> | NSR | 75.7 ± 0.9 | 82.4 ± 0.9 | 87.0 ± 0.8 | 90.1 ± 0.7 | **92.4 ± 0.6** | **94.1 ± 0.6** | **95.3 ± 0.5** | **96.2 ± 0.5** | **96.9 ± 0.5** |
> | W-REINFORCE | **76.6 ± 0.9** | **82.8 ± 0.9** | **87.1 ± 0.8** | **90.2 ± 0.7** | **92.4 ± 0.6** | **94.1 ± 0.6** | **95.3 ± 0.5** | 96.1 ± 0.5 | 96.7 ± 0.5 |
>
> As shown in the table, NSR/W-REINFORCE consistently matches or outperforms the state-of-the-art RL algorithms PPO and GRPO across a wide range of pass@k values. Notably, for $k > 4$, both W-REINFORCE and NSR achieve statistically significantly higher pass@k scores than GRPO. We'd like to highlight that W-REINFORCE achieves this competitive performance with a much simpler design: no value model (unlike PPO) or sophisticated group-wise advantage normalization (unlike GRPO). Given these advantages, we believe our method offers a good trade-off between performance and simplicity, and can serve as a strong baseline for future research.
>
> > **The authors only tested one model (quality, uncertain significance)...**
>
> To address your concern, we conduct experiments under the same setting with a different model family. Following are results with Llama-3.1-8B-Instruct as the backbone model on MATH:
>
> | Method | 1        | 2        | 4        | 8        | 16       | 32       | 64       | 128      | 256      |
> | ------ | -------- | -------- | -------- | -------- | -------- | -------- | -------- | -------- | -------- |
> | PPO    | 52.0     | 59.9     | 66.3     | 71.7     | 76.1     | 79.6     | 82.6     | 85.2     | 87.4     |
> | GRPO   | **54.1** | 61.7     | **67.7**     | 72.6     | 76.7     | 80.1     | 83.0     | 85.5     | 87.5     |
> | PSR    | 50.2     | 56.3     | 61.7     | 66.3     | 70.4     | 73.9     | 76.9     | 79.6     | 81.8     |
> | NSR    | 52.8     | **60.8** | 67.4 | **72.9** | **77.4** | **81.2** | **84.3** | **86.7** | **88.7** |
>
> As shown in the table, NSR is also effective and comparable or outperforms other methods with Llama model.
>
> > **What are the implications of splitting rewards into “only positive” and “only negative”... does encountering a positive sample every 10 negative samples impact the overall trajectory differently when comparing trajectories with only positive or negative samples?**
>
> The positive and negative rewards in RLVR are typically entangled and difficult to interpret directly, which motivates our decomposition into “only positive” and “only negative” rewards. This controlled setup enables us to isolate and understand the independent effects of each reward signal. As shown in Figure 3, we present the training trajectories for three scenarios: both positive and negative rewards (PPO/GRPO), only positive rewards (PSR), and only negative rewards (NSR).
>
> We believe PPO and GRPO’s trajectories reflect the practical training dynamics of mixed rewards because they are designed to handle varying ratios of correct and incorrect rollouts using advantage normalization to adapt to the natural distribution of rewards. We do not study the scenario with a manually fixed positive-to-negative ratio (e.g., 1:10) as it introduces a sampling bias that deviates from the actual training dynamic and is not a realistic representation of on-policy learning.
>
> > **To what extent are the results in Figure 2 explained by PPO and GRPO training encountering more negative than positive reward samples during training?**
>
> We clarify that PPO and GRPO do not encounter more negative than positive reward samples. Instead, as shown in Figure 3(c), PPO and GRPO encounter more positive than negative reward samples during training. This leads to increasing probability of generating observed correct samples and causes a reduction in model entropy, as shown in Figure 3(b). The narrowing of the output distribution limits the model’s exploration, which explains the inferior  pass@k performance for larger k shown in Figure 2.
>
> > **To what extent is the trend of RL models losing their advantage at larger decoding budgets solely explained by negative reward samples (as argued in lines 136-141)? Are there no other possible explanations?**
>
> The major difference between NSR and GRPO/PPO is that NSR relies solely on negative reward samples while excluding positive ones, while PSR represents the opposite extreme. This tightly controlled setup allows us to directly attribute the observed trends to the presence or absence of positive and negative rewards.
>
> While we cannot claim with absolute certainty that no other factors contribute in any way, our controlled experimental design is a standard scientific method for isolating a primary cause (i.e., positive reward only training yields pass@k degradation). The fact that we can (a) turn the degradation effect on and off by manipulating a single variable (the positive reward signal) and (b) provide a clear, low-level mechanistic explanation for it (the gradient dynamics) makes our explanation robust and direct, minimizing the likelihood of other major confounding factors.
>
> > **It is unclear to me if the authors agree with this statement “[line 265] RLVR primarily adjusts the model’s output distribution toward high-reward responses, rather than eliciting new reasoning abilities beyond the base model.”**
>
> Whether RLVR elicits new reasoning abilities or primarily adjusts the output distribution depends on the specific setup and task, and generally there isn’t a clear-cut answer. Our paper does not aim to resolve this question. Instead, our focus is to highlight a previously under-appreciated role of the negative sample reinforcement in shaping model behavior in RLVR. An in-depth investigation into the emergence of reasoning ability is orthogonal to our contributions and beyond the scope of this work.
>
> > **What can we learn or hypothesize from your study about RL algorithms using dense/progress-based rewards?**
>
> In dense reward cases, where the reward may be in the range of -1~1, the findings of our paper can still transfer to such a scenario: positive reward will reinforce observed rollouts, negative reward will promote other possible candidates.

---

> > ### Comment · Reviewer_hPq3 · 2025-08-06
> >
> > Thank you for your very detailed rebuttal, additional experiments, and overall clarifications. Indeed, the rebuttal addresses all of my main concerns and I think this paper would be a great addition to NeurIPS 2025. I will increase my score to a 5 and ask the authors to include the rebuttal answers to my raised concerns in the final version of the paper.
> >
> > Also, I wanted to get the author's feedback on this point I made in the original review (this will not impact my review score):
> >
> > - _"Understanding the importance of negative samples can also have a significant impact on alignment and safety methodologies, where constructing negative samples based on a rule or reason is easier than constructing positively rewarded ones."_
> >
> > Do you think the statement is true and whether your findings could improve either the generalization or robustness of RLHF-based alignment methods of LMs (which usually use PPO and GRPO)? If yes, this could be worth mentioning in the paper or an appendix.

---

> > > ### Author Response · Authors · 2025-08-08
> > >
> > > Thank you for raising the score. We appreciate the encouraging words and the insightful question. Regarding the follow-up question: Yes, we believe your statement is true and our findings can generalize to RLHF-based alignment methods. We agree that in real-world applications, it is often easier to collect negative samples from model failures and bad cases than to curate high-quality positive samples. Based on our findings, performing negative sample reinforcement on these negative examples can discourage the model from generating similar content and drive the model toward other plausible and safer behaviors.
> > >
> > > We will incorporate the discussions above in the revision. Thanks again for your time!

---

> ### Author Response · Authors · 2025-08-06
> **Please take a moment to review our rebuttal**
>
> Dear Reviewer,
>
> As the discussion deadline approaches, we kindly request that you take a moment to review our rebuttal. Thank you again for your thoughtful feedback.

---

### Official Review · Reviewer_CHWn · 2025-07-03

**Clarity:** 3
**Significance:** 2
**Originality:** 3
**Rating:** 4
**Confidence:** 4

**Summary:**

The paper decomposed the training signals of reinforcement learning into two independent parts: "rewarding correct answers" (positive reinforcement, PSR) and "punishing incorrect answers" (negative reinforcement, NSR), and made an unexpected key discovery. The study shows that merely training the model by punishing incorrect answers (NSR) is sufficient to match or even surpass more complex algorithms like PPO, because it can enhance the reasoning ability while effectively protecting the output diversity of the model, which is crucial for complex problems that require multiple attempts to solve. On the contrary, merely rewarding correct answers (PSR) can improve the single-shot accuracy rate, but it will damage diversity due to overconfidence. Based on this, the paper reveals that negative reinforcement works by gently correcting the model's prior knowledge, and proposes a simple and efficient W-REINFORCE algorithm, which achieves excellent results in multiple mathematical benchmark tests by effectively balancing positive and negative feedback.

**Questions:**

1.The new algorithm W-REINFORCE proposed in the paper sets the weight of positive samples to λ = 0.1. The experimental results show that this setting is very effective. However, the paper does not seem to provide an ablation study on the selection of this hyperparameter. How was λ selected? Is it sensitive to different models and tasks? What would the results be if λ is set to 0.05 or 0.2? A robust method should be less sensitive to the hyperparameters, or at least should clearly demonstrate its sensitivity.

2.In the experiment, only the Qwen2.5-Math-7B and Qwen3-4B models were used for the experiment here. It is unknown whether this method is applicable in different model sizes as well.

3.The paper, in the appendix, acknowledges that long-term training of NSR leads to performance degradation. However, this should not merely be regarded as a "limitation", but rather a core experimental result that deserves in-depth analysis in the main text. Readers cannot discern the "decline" trend of NSR from the charts in the main text because the experiment may only have reached its peak performance. The stability of an algorithm is a crucial aspect in evaluating its value.

**Ethical Concerns:**

["NO or VERY MINOR ethics concerns only"]

**Final Justification:**

The author's detailed reply addressed most of my concerns. So I raise my rating. The additionally conducted experiments demonstrating the effectiveness of NSR on different model families and the robustness of W-REINFORCE.

**Limitations:**

Yes

**Quality:**

3

**Strengths And Weaknesses:**

Strengths:
The paper decomposes the training signals of reinforcement learning into two independent parts: "rewarding correct answers" (positive reinforcement, PSR) and "punishing incorrect answers" (negative reinforcement, NSR). Its most significant finding is that "only punishing errors" can achieve excellent results, which challenges the common perception that people have previously placed more emphasis on rewarding correct answers, and reveals the huge value of negative feedback that has been underestimated.

Weaknesses：
1.The conclusion of the paper heavily relies on the Qwen-Math type of "expert models" that are already extremely powerful. The reason why negative reinforcement works is that the model already has the ability to distinguish "suboptimal solutions". If it were replaced with an ordinary, non-professional basic model, the effectiveness of this method would likely be greatly reduced, and thus its universality is questionable.
2.This method is highly suitable for tasks in mathematics that have clear, binary (right or wrong) feedback signals. However, it is difficult to apply to tasks such as creative writing, dialogue systems, etc., where the reward signals are ambiguous, subjective or continuous. This is because it is impossible to clearly distinguish between "positive" and "negative" samples.
3.Long-term use of negative reinforcement alone leads to performance degradation, indicating that it is not a stable training strategy in the long run. Additionally, the new algorithm W-REINFORCE proposed by them lacks sufficient ablation analysis for the selection of key hyperparameters.

---

> ### Author Rebuttal · Authors · 2025-07-31
>
> Thanks for your thoughtful feedback. We address your raised issues as follows.
>
> > **The conclusion of the paper heavily relies on the Qwen-Math type of "expert models" that are already extremely powerful… and thus its universality is questionable.**
>
> To address your concern, we conduct experiments under the same setting with a different model family. Following are Pass@k results with Llama-3.1-8B-Instruct as the backbone model on MATH
>
> | Method | 1        | 2        | 4        | 8        | 16       | 32       | 64       | 128      | 256      |
> | ------ | -------- | -------- | -------- | -------- | -------- | -------- | -------- | -------- | -------- |
> | PPO    |  52.0     |   59.9     |   66.3     | 71.7     | 76.1     | 79.6     | 82.6     | 85.2     | 87.4     |
> | GRPO | **54.1** | **61.7** | **67.7** | 72.6     | 76.7     | 80.1     | 83.0     | 85.5     | 87.5     |
> | PSR    |   50.2     |   56.3   |     61.7    | 66.3     | 70.4     | 73.9     | 76.9     | 79.6     | 81.8     |
> | NSR    |   52.8     |   60.8   |   67.4    | **72.9** | **77.4** | **81.2** | **84.3** | **86.7** | **88.7** |
>
> As shown in the table, NSR is also effective and comparable or outperforms other methods with Llama model.
>
> > **This method is highly suitable for tasks in mathematics that have clear, binary (right or wrong) feedback signals. However, it is difficult to apply to tasks such as creative writing, dialogue systems, etc., where the reward signals are ambiguous, subjective or continuous. This is because it is impossible to clearly distinguish between "positive" and "negative" samples.**
>
> As stated in the paper, our focus is on reinforcement learning with **verifiable rewards** (RLVR), where the reward is inherently unambiguous and verifiable. This setup aligns with prior RLVR literature that uses binary rewards [1,2,3,4]. Open-ended tasks like creative writing and dialogue are typically addressed using RLHF frameworks [5] which represent a different training setting (e.g., Qwen 3 [4] has separate training stages for verifiable and non-verifiable tasks). Therefore, extending our analysis and method to non-verifiable tasks is an interesting direction for future work, but is beyond the scope of RLVR.
>
> > **Long-term use of negative reinforcement alone leads to performance degradation, indicating that it is not a stable training strategy in the long run. Additionally, the new algorithm W-REINFORCE proposed by them lacks sufficient ablation analysis for the selection of key hyperparameters.**
>
> We explicitly acknowledged this training stability observation in our paper to promote transparency and to encourage further research in RLVR. That said, we also note that training instability or performance degradation over extended RL training is a well-known challenge in the broader RL-for-LLM-reasoning research, not unique to our method. For example, standard RL methods (e.g., PPO) also suffer from training instability or collapse over long horizons for LLM training, as shown in Figure 2 of [2].
>
> As for hyperparameter of W-REINFORCE, we provide ablations with different λ as follows:
>
> MATH (Pass@k)
>
> | λ    | 1        | 2        | 4        | 8        | 16       | 32       | 64       | 128      | 256      |
> | ---- | -------- | -------- | -------- | -------- | -------- | -------- | -------- | -------- | -------- |
> | 0.05 | 75.6     | 82.2     | 86.7     | 89.9     | 92.2     | 94.0     | 95.0     | **96.3** | **97.1** |
> | 0.1  | **76.6** | **82.8** | **87.1** | **90.2** | **92.4** | **94.1** | **95.3** | 96.1     | 96.7     |
> | 0.2  | 75.8     | 81.2     | 85.2     | 88.3     | 90.7     | 92.6     | 94.0     | 95.1     | 95.9     |
>
> As shown in the above table, we test λ=0.05, 0.1, 0.2, large λ makes the method closer to PSR and small λ makes it closer to NSR. Results show that W-REINFORCE is robust with different λ.
>
> [1] DeepSeek-R1: Incentivizing Reasoning Capability in LLMs via Reinforcement Learning, 2025
> [2] Demystifying Long Chain-of-Thought Reasoning in LLMs, 2025
> [3] SimpleRL-Zoo: Investigating and Taming Zero Reinforcement Learning for Open Base Models in the Wild, 2025
> [4] Qwen3 Technical Report, 2025
> [5] Training language models to follow instructions with human feedback, 2022

---

> > ### Comment · Reviewer_CHWn · 2025-08-07
> >
> > Thanks for authors's detailed response, which solved my main questions. So I am willing to raise my score.
> >
> > I'd like to discuss one more question. How do you think the the NSR (now in RLVR) methods perform in not verified tasks as mentioned in weakness2? Or does it still depend on model priors, as you replied to the other reviewer?

---

> > > ### Author Response · Authors · 2025-08-08
> > >
> > > Thank you for raising the score and the insightful question. We discuss two factors that affect the effectiveness of NSR in non-verifiable tasks.
> > >
> > > - **Reward model (RM) quality**: For subjective tasks, an RM provides the essential signal for what constitutes a "good" or "bad" response. The reward signal shifts from verifying correctness (in RLVR) to acting as a proxy for desired qualities like helpfulness, creativity, or safety. As long as the RM faithfully captures the target preferences, the analysis and findings in our paper will hold, as NSR’s mechanism effectively translates negative reward signals into policy updates.
> > > - **Dependence on model prior**: You are correct that the model's priors are still critical in non-verifiable tasks. As our gradient analysis reveals, NSR penalizes mistakes by reallocating probability mass from negative-reward responses to alternatives that the current policy already prefers. In non-verifiable tasks, a strong prior means the model already possesses the necessary capabilities (e.g., stylistic variety, relevant knowledge) to generate high-quality candidates. NSR then acts as a refiner, using the RM's signal to prune undesirable paths while strengthening the high-quality behaviors already latent within the model.
> > >
> > > In essence, for subjective tasks, the RM provides the directional signal, while the model's prior determines how efficiently the model can realize it with NSR. We will incorporate the discussions above in the revision. Thanks again for your time!

---

> ### Author Response · Authors · 2025-08-06
> **Please take a moment to review our rebuttal**
>
> Dear Reviewer,
>
> As the discussion deadline approaches, we kindly request that you take a moment to review our rebuttal. Thank you again for your thoughtful feedback.

---

### Official Review · Reviewer_Mh3W · 2025-07-14

**Clarity:** 3
**Significance:** 2
**Originality:** 2
**Rating:** 2
**Confidence:** 2

**Summary:**

This paper study the effect of the gradient of positive and negative samples on the policy. It conduct an experiment to show the influence of tdifferent gradient on different Pass@k accuracy. It then design an algorithm to improve Pass@k accuracy by giving different weight to different part of the loss and gradient.

**Questions:**

Why doesn't consider the setting of off-policy?

**Ethical Concerns:**

["NO or VERY MINOR ethics concerns only"]

**Limitations:**

1. The paper primarily focuses on the on-policy setting. However, off-policy settings—such as those used in popular algorithms like DPO—may exhibit different behaviors. The lack of consideration for off-policy scenarios limits the generality and significance of the findings.

**Quality:**

2

**Strengths And Weaknesses:**

Strengths:
The paper is clearly written, and the experiments effectively demonstrate the main observations.

Weaknesses:

1. Personally, I find that the phenomenon observed is somewhat aligned with common intuition in the field. Additionally, the technique of reweighting the negative samples lacks novelty in my view. However, I acknowledge that other reviewers may differ in their judgment on whether this insight constitutes a meaningful contribution beyond what is already understood without formal validation.

2. The paper primarily focuses on the on-policy setting. However, off-policy settings—such as those used in popular algorithms like DPO—may exhibit different behaviors. The lack of consideration for off-policy scenarios limits the generality and significance of the findings.

---

> ### Author Rebuttal · Authors · 2025-07-31
>
> Thanks for your thoughtful feedback. We address your raised issues as follows.
> > **Personally, I find that the phenomenon observed is somewhat aligned with common intuition in the field. Additionally, the technique of reweighting the negative samples lacks novelty in my view.**
>
> To the best of our knowledge, our work is the first to find that solely penalizing mistakes through negative reinforcement learning, without relying on positive reward signals, can effectively improve LLM reasoning. If we missed any relevant work, we are happy to discuss and cite them accordingly.
>
>
> > **The paper primarily focuses on the on-policy setting. However, off-policy settings—such as those used in popular algorithms like DPO—may exhibit different behaviors. The lack of consideration for off-policy scenarios limits the generality and significance of the findings.**
>
> On-policy RL training has been increasingly used in the latest state-of-the-art models. Notably, models such as DeepSeek-R1 [1], Kimi [2], and Qwen3 [3] all have adopted **fully on-policy training** without any off-policy RL components. Moreover, recent RLVR research [4,5] predominantly adopts on-policy RL algorithms like PPO and GRPO. Therefore, we believe our findings under the on-policy setting are both significant and relevant to current practice.
>
> [1] DeepSeek-R1: Incentivizing Reasoning Capability in LLMs via Reinforcement Learning, 2025
> [2] Kimi k1.5: Scaling Reinforcement Learning with LLMs, 2025
> [3] Qwen3 Technical Report, 2025
> [4] Demystifying Long Chain-of-Thought Reasoning in LLMs, 2025
> [5] SimpleRL-Zoo: Investigating and Taming Zero Reinforcement Learning for Open Base Models in the Wild, 2025

---

> ### Author Response · Authors · 2025-08-06
> **Please take a moment to review our rebuttal**
>
> Dear Reviewer,
>
> As the discussion deadline approaches, we kindly request that you take a moment to review our rebuttal. Thank you again for your thoughtful feedback.

---

> ### Author Response · Authors · 2025-08-08
> **Discussion Deadline Reminder**
>
> Dear Reviewer,
>
> As the discussion deadline (August 8, AOE) is approaching, we kindly remind you to take a moment to review our rebuttal. We sincerely appreciate your time and feedback.

---

### Comment · Area_Chair_rJSj · 2025-08-06

Dear reviewers,

This is a reminder that the end of author-reviewer discussion period is **Aug. 8**. Please do carefully read all other reviews and the author responses; and discuss openly with the authors, especially on your own questions that the authors addressed.

Best,
AC

---

### Decision · Program_Chairs · 2025-09-17

**Decision:**

Accept (poster)

**Comment:**

The paper studies reinforcement learning with verifiable rewards by decomposing signals into positive (PSR) and negative (NSR) reinforcement. It finds that NSR alone can substantially improve LLM reasoning, often matching or surpassing PPO/GRPO, while preserving diversity. The authors propose W-REINFORCE, a simple variant emphasizing NSR, which performs competitively across reasoning benchmarks (MATH, AIME 2025, AMC23).

**Strengths:**

- Clear and well-structured, with strong empirical evidence.

- Provides novel insight into the value of negative reinforcement in LLM training.

- Demonstrates improved diversity and robustness compared to PSR-only methods.

- W-REINFORCE is simple yet effective, rivaling more complex baselines.

- Rebuttal added significance tests, additional model experiments, and ablations.

**Weaknesses:**

- Generality limited to strong base models and binary/verifiable tasks.

- NSR alone degrades with long-term training.

- Hyperparameter sensitivity and novelty relative to concurrent work remain mild concerns.

--

Reviewers initially raised concerns on statistical rigor, model diversity, stability, novelty, and $\lambda$ sensitivity. The authors provided confidence intervals, tested on a second model family (Llama), added $\lambda$ ablations, and clarified scope and distinctions from concurrent work. These responses resolved most concerns, leading to higher scores and a clear positive consensus.

The key contribution (that penalizing mistakes alone can improve reasoning) is interesting. Rebuttal experiments convincingly addressed statistical, robustness, and novelty concerns. Three reviewers moved to acceptance; only one remained negative with limited engagement. On balance, this is a solid contribution meriting acceptance.